# Machine Learning Facilitates Hotspot Classification in PSMA-PET/CT with Nuclear Medicine Specialist Accuracy

**DOI:** 10.3390/diagnostics10090622

**Published:** 2020-08-22

**Authors:** Sobhan Moazemi, Zain Khurshid, Annette Erle, Susanne Lütje, Markus Essler, Thomas Schultz, Ralph A. Bundschuh

**Affiliations:** 1Department of Nuclear Medicine, University Hospital Bonn, 53127 Bonn, Germany; annette.erle@t-online.de (A.E.); Susanne.Luetje@ukbonn.de (S.L.); markus.essler@ukbonn.de (M.E.); ralph.bundschuh@ukbonn.de (R.A.B.); 2Department of Computer Science, University of Bonn, 53115 Bonn, Germany; schultz@cs.uni-bonn.de; 3Department of Nuclear Medicine, Nuclear Medicine, Oncology and Radiotherapy Institute, 21061 Islamabad, Pakistan; dr.zainkhurshid@gmail.com; 4Bonn-Aachen International Center for Information Technology (B-IT), University of Bonn, 53115 Bonn, Germany

**Keywords:** prostate cancer (PC), prostate-specific membrane antigen (PSMA), positron emission tomography (PET), computed tomography (CT), machine learning (ML)

## Abstract

Gallium-68 prostate-specific membrane antigen positron emission tomography **(**^68^Ga-PSMA-PET) is a highly sensitive method to detect prostate cancer (PC) metastases. Visual discrimination between malignant and physiologic/unspecific tracer accumulation by a nuclear medicine (NM) specialist is essential for image interpretation. In the future, automated machine learning (ML)-based tools will assist physicians in image analysis. The aim of this work was to develop a tool for analysis of ^68^Ga-PSMA-PET images and to compare its efficacy to that of human readers. Five different ML methods were compared and tested on multiple positron emission tomography/computed tomography (PET/CT) data-sets. Forty textural features extracted from both PET- and low-dose CT data were analyzed. In total, 2419 hotspots from 72 patients were included. Comparing results from human readers to those of ML-based analyses, up to 98% area under the curve (AUC), 94% sensitivity (SE), and 89% specificity (SP) were achieved. Interestingly, textural features assessed in native low-dose CT increased the accuracy significantly. Thus, ML based on ^68^Ga-PSMA-PET/CT radiomics features can classify hotspots with high precision, comparable to that of experienced NM physicians. Additionally, the superiority of multimodal ML-based analysis considering all PET and low-dose CT features was shown. Morphological features seemed to be of special additional importance even though they were extracted from native low-dose CTs.

## 1. Introduction

Computer-aided diagnosis (CAD) based on artificial intelligence (AI) and machine learning (ML) will revolutionize the process of image reading in radiology and nuclear medicine [1]. Innovative tools will assist physicians in handling large data-sets of images more efficiently. A central issue in this context will be the development of tools for the automated classification of lesions to pre-define pathological findings following work-up by the physician. CAD was proposed as early as 1998 for lung nodules in computed tomography (CT) examinations [2]. To date, many other applications have been described—for example, in mammography [3] and positron emission tomography (PET) [4]. More recently, radiomics features such as textural parameters became the focus of interest in the analysis of imaging data in PET as well as in CT or magnetic resonance imaging (MRI). The significance of textural features analysis in diagnosis and therapy response prediction using PET/CT has been demonstrated by a large body of evidence [5,6,7,8,9,10,11].

In recent years, radiolabeled analogues of the prostate-specific membrane antigen (PSMA) were developed for imaging of primary prostate cancer (PC) and PC metastases. Gallium-68 (^68^Ga) and Fluorine-18 (^18^F)-labeled PSMA tracers are highly effective and show high detection rates, especially in PC patients with biochemical recurrence [12,13,14]. Due to its high sensitivity, PSMA-PET/CT helps to stratify patients in primary staging of PC for surgery or for systemic treatment by exclusion or detection of metastases [15,16]. Therefore, PSMA-PET/CT has become the most important imaging modality, especially for staging and restaging of PC. Thus, it is frequently performed at most comprehensive cancer centers. However, for optimal therapy decisions, accurate scan interpretation is essential [17] to guide the referring physician to handle challenging cases or to recommend appropriate work-up. For standardization of reporting and reduction of reporting time, it would be of high interest to develop AI-based tools for automatic discrimination of malignant lesions from physiological PSMA uptake.

Here, we report an innovative in-house programmed tool including five different ML algorithms for the classification of lesions in PSMA-PET/CT images as pathological or physiological based on analysis of textural features. Our data collective consists of 72 patients with 2419 PSMA-positive findings. By means of the tool, it is possible to discriminate unspecific from malignant PSMA tracer accumulations with similar sensitivity and specificity to trained nuclear medicine physicians.

## 2. Materials and Methods

### 2.1. Patients and Volume of Interest (VoI) Definition and Annotation

In total, 72 male patients with histologically confirmed prostate carcinoma who were referred for ^68^Ga-PSMA PET/CT were included in this retrospective analysis. The patients′ ages ranged from 48 to 87 years, Gleason score ranged from 6 to 9 and serum PSA level from 4.0 ng/mL to 1840 ng/mL. All patients had undergone previous treatments: 63 underwent radical prostatectomy, 11 received local radiation treatment of the prostate, 69 had hormonal treatment, 56 received chemotherapy, and 47 underwent radiation treatment of bone or lymph node metastases. All patients were referred to our department for follow-up staging, with the possibility of further nuclear medicine treatment either with radium-223-dichloride or luthetium-177-PSMA. The scans were carried out between November 2014 and February 2017 using a Biograph 2 PET/CT system (Siemens Medical Solutions, Erlangen, Germany). Around 40 to 80 min after intravenous injection of 98 to 159 MBq in-house produced ^68^GA-HBED-CC PSMA, a low-dose CT (16mAs, 130 kV) from the base of skull to mid-thigh was acquired, followed by the PET scan acquired over the same area, with 3 or 4 min per bed position depending on the body weight of the patient. CT data were reconstructed in 512 to 512 matrices with 5 mm slice thickness. PET data were reconstructed in 128 by 128 matrices with 5 mm slice thickness as well. An attenuation-weighted ordered subsets expectation maximization algorithm was utilized for image reconstruction, including attenuation and scatter corrections as implemented by the manufacturer. Written informed consent to the imaging procedure and for anonymized evaluation of their data was obtained from all patients. Due to the retrospective character of the data analysis, an ethical statement was waived by the institutional review board.

For each scan, all the hotspots have been identified and manually delineated consecutively by two trained nuclear medicine (NM) physicians (both board-certified and with 7 and 3 years’ experience in PET/CT) using InterView FUSION software V3.08.005 (Mediso Medical Imaging Systems, Budapest, Hungary [18]) (see Figure 1). Hotspots were defined as focal uptake beyond the local background without any specific threshold. To define each 3D hotspot, all its 2D counterparts were delineated in subsequent slices. Hence, the hotspots were analyzed as fully connected 3D volumes. The hotspots included malignant tissues in any organs and metastatic uptakes in bones or lymph nodes as well as physiological uptakes in kidneys, livers, etc., as well as benign or unspecific uptake, e.g., in the thyroid. Per hotspot, a total of 80 (40 PET-based+40 CT-based) features were calculated by InterView FUSION software (the standard set of radiomics features provided by the software). The features include first and higher order statistics features (mean, max, kurtosis, etc.), shape-based features (max diameter and volume), textural features (entropy, contrast, homogeneity, etc.), and volumetric zone and run length statistics (grey-level non-uniformity, short run emphasis, etc.). See Table 1 for a detailed list of the radiomics features. Afterwards, the ground truth labels were merged with the features calculated by InterView software using our internal PET/CT scan annotator software (Python V2.7).

### 2.2. Classification

After delineation, as the ground truth labels, the hotspots were divided into two classes by two experienced NM physicians: pathological (malignant) vs. physiological (unspecific). After accumulating the data from all the scans, the feature vector was divided into three feature groups: PET only, CT only, and combined PET and CT (PET/CT). Five different ML algorithms (linear, radial basis function (RBF), and polynomial kernel support vector machine (SVM) [19], extra trees (ET) [20], and random forest (RF) [21]) were applied to each subset of the features. Hence, the performance of all classifiers was quantified as applied to each of the feature groups (e.g., PET with ET or PET/CT with linear SVM).

To quantify the significance of our results, the accuracy measures (area under the receiver operating characteristic (ROC) curve (AUC), standard deviation (STD) of AUCs for the cross-validation step as well as for the feature importance, sensitivity (SE), and specificity (SP)) were quantified to calculate the total precision for each of the classifiers applied to each feature group. Hyperparameter values for the ML methods were established using five-fold cross-validation (CV) on the training data-set with 48 subjects. The performance of the resulting classifier was evaluated on a validation (hold-out) set with 24 subjects followed by an inter-observer analysis.

To gain insight into which features contributed most, we also ranked them based on the ET classifier, as it performed best. The features were ranked with the ET feature importance measure provided by the scikit learn library [22]. It quantifies the overall decrease in Gini impurity achieved with a given feature. We report means and standard deviations of these importance scores, over the folds of our five-fold cross-validation.

### 2.3. Cross-Validation (CV)

To achieve more generalizable results, it is important to use separate data for tuning model hyperparameters and for evaluating the final accuracy. We thus randomly sub-divided our data into two subsets. The first subset (named the training set and including 48 subjects) was used for training and hyperparameter tuning using cross-validation. The second subset, containing 24 subjects, was used as the validation or hold-out set. After standardizing the data-set using the MinMaxScaler method [23], cross-validation using the KFold method with five folds was applied on the training set. In each CV step, a grid search was performed to find the best set of parameters for the ML algorithms to predict the true labels for each category. For the grid search, all the five ML classifiers (SVM with three different kernels including linear, polynomial, and RBF, as well as random forest and extra trees) were tested with different parameters (C = [1, 10, 100, 1000, 2^−5^, 2^−3^,..., 2^15^], gamma = [10^−3^, 10^−4^, 2^−15^, 2^−13^, 2^−11^,..., 2^3^], etc.).

Given the best set of parameters for each classifier on the training set, the performance of each classifier to predict the labels of the validation set was calculated. Again, the relative importance of each feature group was calculated individually. We report the accuracy measures of each classifier on each feature group applied to the validation (hold-out) set.

### 2.4. Inter-Observer Variability

To check for inter-observer variability, both qualitative and quantitative measures were taken into account. The whole cohort was randomly divided into two subsets: one with 30 patients and one with 42 patients. Each subset was annotated and manually segmented by a different experienced NM physician (hence, two annotators)—both board-certified and one with 7 years′ experience in PET/CT and the other with 3 years′ experience in PET/CT. Due to availability of the retrospective data as well as different time limits, the NM physicians were assigned to annotate different numbers of patients. However, we made sure that the cohorts had similar demographic and physiological distributions. Afterwards, the segmentation results by the two annotators were reviewed and qualified by a third highly experienced NM physician (also board-certified with 7 years′ experience in PET/CT). To quantify the variability of the manual segmentations, additional rounds of CV were applied. In the first round, training data came from the first subset and test data came from the second subset. In the second round, the train and test data swapped sides. The results were then compared to the main CV results.

### 2.5. Permutation Test

At the end, a permutation test was performed to reject the null hypothesis which stated that permuted distribution of labels might have produced similar results. Here, we conducted a separate five-fold CV on the cohort with 48 patients from the first CV step. There were 25,000 total iterations with the same set of feature groups and ML classifiers. In each CV step, the ground truth labels were replaced with permuted binary labels. We counted each prediction score (AUC) equal to or higher than the threshold of 0.85 (which was lower than our worst prediction score on the hold-out set). Then, we divided the resulting number by the total number of iterations (25,000) to calculate the *p*-value of the permutation test:(1)$$p\;=\frac{n\;\left(AUCs\geq \;0.85\right)}{N_{iters}}$$
where p is the *p*-value of the permutation test, n () is the number of the test scores over the given threshold, AUCs are the calculated areas under the ROC curves for each classifier on each feature group at each iteration, and $N_{iters}$ is the total number of iterations (Equation (1)).

## 3. Results

First, 2419 focal tracer accumulations were delineated manually throughout the collective of 72 PC patients. Out of these lesions, 1629 were classified as pathological and 790 as physiological. Table 2 illustrates the distribution of the hotspots throughout different body regions. Based on these data, the five ML algorithms were applied to the 48 training set patients. The training set patients were randomly selected from the main cohort. Each ML algorithm was applied on all subsets of data (PET only, CT only, and PET/CT). The results of this first train and test step using five-fold CV are shown in Figure 2 and Table 3. As shown, highly accurate classification scores (up to 98% AUC, 96% SE, 91% SP) were achieved. Interestingly, the contribution of the CT-based features to the results becomes apparent by these data.

To avoid over-fitting, a second validation step was taken. For this purpose, the remaining 24 patients were used as the validation set. As shown in Figure 3 and Table 4, accuracy measures increase when comparing PET with CT and PET/CT (up to 98% AUC, 94% SE, 89% SP). Again, the CT feature group showed surprisingly good results. Amongst the different ML algorithms, decision tree-based classifiers (RF and ET) showed the best performance, regardless of the subset used.

To test the stability of our data upon delineation by different nuclear medicine specialists, the inter-observer variability of the accuracy measures (AUC, SE, and SP) obtained by the different algorithms was determined. We found that delineation by different observers does not markedly change the AUCs and sensitivities. Therefore, the random forest algorithm was the most stable method. However, specificity is lower compared to the same measures from CV or the final validation steps (see Figure 4). Figure 5 shows the ranking of the 20 most contributing features regarding the extra trees classifier to predict malignant vs. unspecific hotspots. Appearance of CT-based features in the highest ranks suggests the importance of the morphological texture for the prediction of malignancy. As expected, PET-based heterogeneity parameters such as kurtosis, busyness, and coarseness play important roles as well. Finally, the permutation test resulted in a *p*-value of 0.00076 after 25,000 iterations of permuted label assignment to the hotspots.

## 4. Discussion

We have shown that ML algorithms are capable of discriminating between malignant or physiological/unspecific tracer accumulations in ^68^Ga-PSMA-PET/CT with similar accuracy as achieved by experienced nuclear medicine physicians. In addition, we identified the most suitable ML algorithm for this application. For this purpose, five different ML methods were compared and tested on multiple PET/CT data-sets. For the analysis, 40 textural features extracted from both PET- and low-dose CT data were used. Altogether, 2419 hotspots in 72 patients were evaluated for malignancy. Our results suggest that the combination of PET- and CT-based features improves the precision of differentiation of malignant from unspecific and physiological tracer accumulations in ^68^Ga-PSMA PET/CT compared to each single modality. However, this finding may not be surprising as the advantage of hybrid imaging compared to single imaging modalities is well known. However, it is still remarkable that the appearance of CT features in the highest ranks for the best classifier (extra trees) suggests that anatomical information provided by CT scans facilitates the detection of malignancy in sites with high tracer uptake, even when only native low-dose CTs were used for analysis. Using fully diagnostic, contrast-enhanced CTs in the future may enhance diagnostic accuracy by means of textural parameters even more. Therefore, further studies should also investigate the benefit of contrast enhancement in this regard. On the other hand, radiation exposure of patients would be increased by fully diagnostic CTs and our data indicate that native low-dose CTs yield good results. Therefore, also the use of textural analysis of low-dose CTs without contrast enhancement should be investigated in other tumor entities.

Amongst the five different ML algorithms, the decision tree-based classifiers (ET and RF) showed the best results. The reason for this finding could be that ET and RF apply feature selection implicitly. Therefore, these ML-based methods are powerful automatic algorithms for the identification of malignant hotspots and should be implemented in further algorithms.

Although this first study analyzed only 72 patients, 2419 hotspots were included in our lesion-based analysis. This number was sufficient to demonstrate high statistical significance in our results. However, larger studies have to be performed in the future. High AUCs and sensitivities were achieved in the inter-observer analyses; however, the relatively low specificity scores indicate the need for studies with more annotators as well as multi-center studies in the future. In addition, beyond the scope of this study was the analysis of how the results can be applied on ^68^Ga-PSMA PET/CT scans with different protocols or obtained with other PET scanners. Moreover, replacing the manual delineation of the tissues with an automated segmentation method would be of further benefit. However, this was beyond the scope of the current study.

As we have shown, ML algorithms help to discriminate between malignant and unspecific findings, but they may also help with decisions for or against certain therapies. It may be possible to design tools for the prediction of therapy response—for example, to ^177^Lu-PSMA therapy in PC patients. As a result, it would be possible to exclude non-responders from this treatment to avoid undesirable side effects and cost-intensive treatments without benefit for the individual patient. In this context, Khurshid et al. reported a significant correlation between some textural parameters such as the mean homogeneity and entropy in ^68^Ga-PSMA-PET scans and response to ^177^Lu-PSMA therapy as determined by the reduction of prostate-specific antigen (PSA) levels [24]. Development of an ML algorithm for therapy decision would be of high interest for general oncology and for the selection of patients for clinical trials and could be an important further step towards individualized tumor therapy.

In the future, it will be desirable to develop ML-based tools with not only equal but superior accuracy compared to nuclear medicine physicians. For this purpose, it will be necessary to compare the results of the human readers as well as the ML-based tool with a gold standard such as histology obtained from biopsies of the lesions in question. However, this will be difficult to achieve as biopsies of multiple sites in each patient are not practicable and are highly questionable from an ethical point of view, especially if we take into account the fact that the mean number of hotspots investigated per patient in this study was 33.6. However, this is an important topic in the field and needs to be addressed in further studies. Furthermore, the presented results and implemented algorithms will be extended to other tumor entities and applied using different PET tracers as well.

## 5. Conclusions

Machine learning based on PET/CT radiomics features can differentiate increased tracer uptake in ^68^Ga-PSMA scans in malignant versus physiological or unspecific changes with high accuracy. This finding is important in the context of automated detection and segmentation for radiomics analysis. The analysis of combined PET and CT radiomics features suggests that they are superior to features estimated in each modality alone, even just using a low-dose CT without intravenous contrast.

## Figures and Tables

**Figure 1 diagnostics-10-00622-f001:**
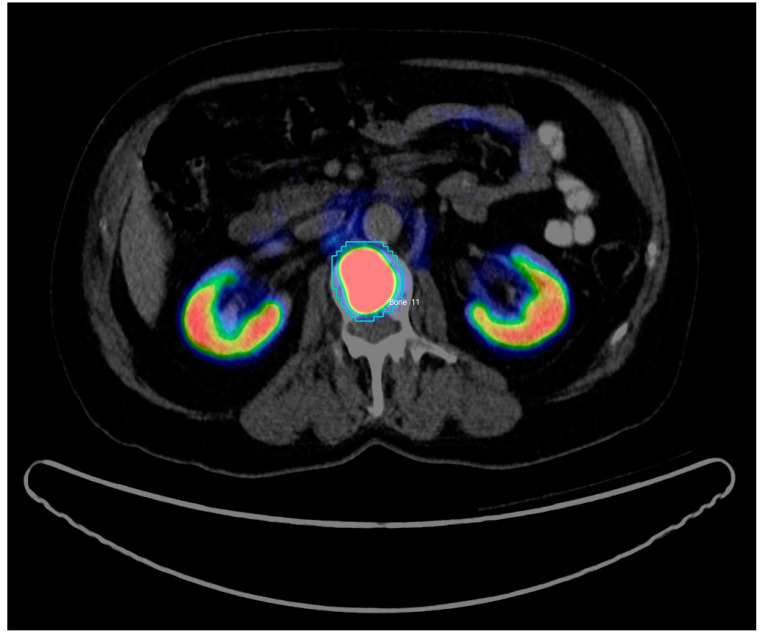
Example of Region of interest (RoI) definition for a bone PET/CT hotspot in InterView FUSION Software. The 2D slice includes the defused PET uptake (kidneys and bone metastasis in dark blue, green, yellow, and red) co-registered with the CT image (in grayscale). The blue contour around the metastatic uptake is defined and named (Bone 11) by the NM expert.

**Figure 2 diagnostics-10-00622-f002:**
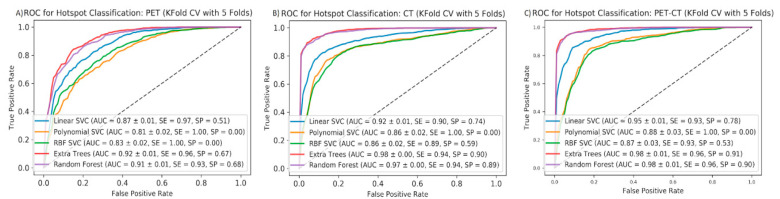
Mean ROC curves for five ML algorithms using five-fold cross-validation to predict pathological vs. non-pathological hotspots using PET (**A**), CT (**B**), and all features (**C**). AUCs (STDs), sensitivities, and specificities are shown for each ML method applied to each feature group.

**Figure 3 diagnostics-10-00622-f003:**
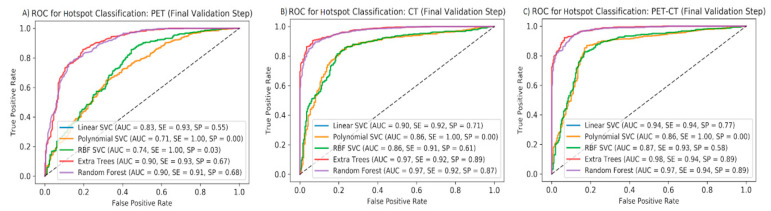
Results of final validation step: ROC curves for five ML algorithms to predict pathological vs. non-pathological hotspots on the validation set using PET (**A**), CT (**B**), and all features (**C**). AUCs, sensitivities, and specificities are shown for each ML method applied to each feature group.

**Figure 4 diagnostics-10-00622-f004:**
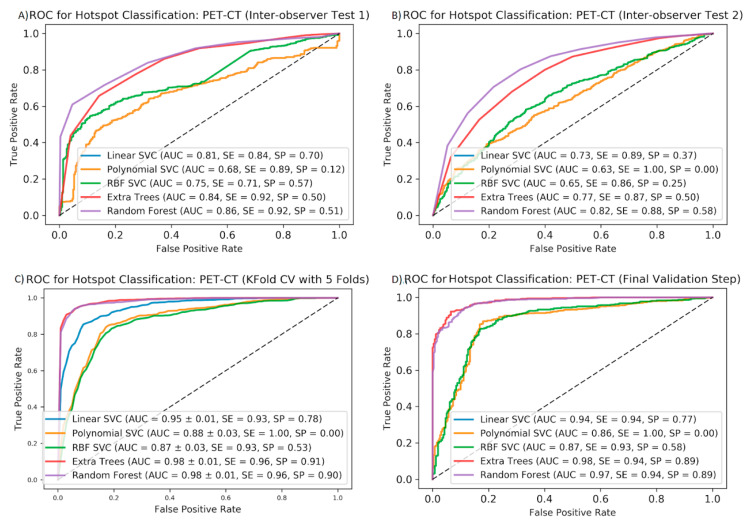
Mean ROC curves for five ML algorithms to predict pathological vs. non-pathological hotspots on the PET/CT features: results of inter-observer test 1 (**A**), inter-observer test 2 (**B**), the five-fold cross-validation (**C**), and the final validation step (**D**). AUCs, sensitivities, and specificities are shown for each ML method in each cross-validation step.

**Figure 5 diagnostics-10-00622-f005:**
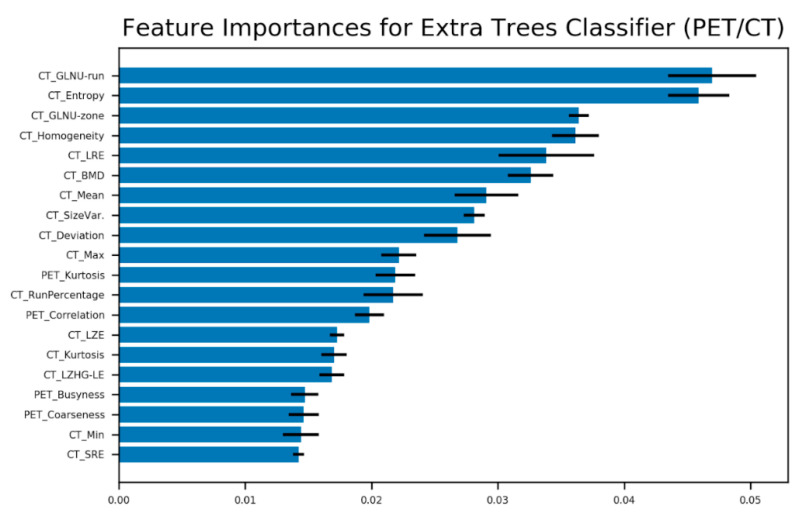
Best 20 features for hotspot classification based on extra trees classifier and five-fold cross-validation. The error bars stand for standard deviation estimated for the CV folds (GreyLevel-NonUniformity (GLNU), LongRunEmphasis (LRE), BoneMineralDensity (BMD), LongZoneEmphasis (LZE), LongZoneHighGrey-LevelEmphasis (LZHG_LE), ShortRunEmphasis (SRE)).

**Table 1 diagnostics-10-00622-t001:** List of the calculated features: PET-based and CT-based features. Note that the total lesion glycolysis (TLG) is PET-specific and BoneMineralDensity is CT-specific.

First or Higher Order Statistics	Shape and Size	Textural	Volumetric Zone Length Statistics	Volumetric Run Length Statistics
Deviation Mean Max Min Sum PET-TLG Kurtosis	Volume Max. Diameter	Entropy Homogeneity Correlation Contrast Size Variation Intensity Variation Coarseness Busyness Complexity CT-BoneMineralDensity	Short Zone Emphasis Long Zone Emphasis Low Grey-Level Zone Emphasis High Grey-Level Zone Emphasis Short Zone Low Grey-Level Emphasis Short Zone High Grey-Level Emphasis Long Zone Low Grey-Level Emphasis Long Zone High Grey-Level Emphasis Grey-Level Non-Uniformity-Zone Zone Length Non-Uniformity Zone Percentage	Short Run Emphasis Long Run Emphasis Low Grey-Level Run Emphasis High Grey-Level Run Emphasis Short Run Low Grey-Level Emphasis Short Run High Grey-Level Emphasis Long Run Low Grey-Level Emphasis Long Run High Grey-Level Emphasis Grey-Level Non-Uniformity-Run Run Length Non-Uniformity Run Percentage

**Table 2 diagnostics-10-00622-t002:** Distribution of the 2419 annotated hotspots over different organs in the 72 patients.

Hotspot Category	Subject Cohorts		Total
	30 Patients	42 Patients	
Metastases	651	969	1620
Bladder	18	40	58
Kidney	59	81	140
Salivary Gland	116	299	415
Others	114	72	186
Total	958	1461	2419

**Table 3 diagnostics-10-00622-t003:** Accuracy measures (area under the curve (AUC), sensitivity (SE), and specificity (SP)) obtained for ML classifiers applied to different feature groups with five-fold cross-validation. The CV cohort contained 48 subjects.

Feature Group	PET	CT	All
**Classifier**	**AUC/SE/SP (%)**	**AUC/SE/SP (%)**	**AUC/SE/SP (%)**
Linear Kernel SVM	87/97/51	92/90/74	95/93/78
Random Forest	91/93/68	97/94/89	98/96/90
Extra Trees	92/96/67	98/94/90	98/96/91
RBF Kernel SVM	83/100/0	86/89/59	87/93/53
Polynomial Kernel SVM	81/100/0	86/100/0	88/100/0

**Table 4 diagnostics-10-00622-t004:** Tuned parameters and accuracy measures (area under the curve (AUC), sensitivity (SE), and specificity (SP)) obtained for ML classifiers applied to different feature groups. The classifiers were trained by the cohort containing 48 subjects and tested on the hold-out set with 24 subjects.

Feature Group	PET		CT		All	
**Classifier**	**Tuned Parameters**	**AUC/SE/SP (%)**	**Tuned Parameters**	**AUC/SE/SP (%)**	**Tuned Parameters**	**AUC/SE/SP (%)**
Linear Kernel SVM	C = 0.5	83/93/55	C = 1	90/92/71	C = 2^11^	94/94/77
Random Forest	max_depth = 30 min_samples_leaf = 1	90/91/68	max_depth = 20 min_samples_leaf = 1	97/92/87	max_depth = 20 min_samples_leaf = 1	97/94/89
Extra Trees	max_depth = 30 min_samples_leaf = 1	90/93/67	max_depth = 10 min_samples_leaf = 1	97/92/89	max_depth = 10 min_samples_leaf = 1	98/94/89
RBF Kernel SVM	C = 2^13^ gamma = 2^−15^	74/100/3	C = 2^−5^ gamma = 2^−15^	86/91/61	C = 2^−3^ gamma = 2^−13^	87/93/58
Polynomial Kernel SVM	C = 1 degree = 2	71/100/0	C = 1 degree = 2	86/100/0	C = 1 degree = 2	86/100/0

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
