# Peer review of "Machine Learning Facilitates Hotspot Classification in PSMA-PET/CT with Nuclear Medicine Specialist Accuracy"

_diagnostics, 2020, doi:10.3390/diagnostics10090622_

Round 1

Reviewer 1 Report

The authors present a very interesting and highly important analysis on ML utilization in hotspot classification of PSMA-PET-CT for prostate cancer. The manuscript is well-written.

Despite its strengths, I recommend adressing the following comments:

  1. Title: The authors rely on a manual segmentation / did not apply a fully automated segmentation. Therefore, I would suggest to rephrase the title in a more cautious way. For example, I would recommend removing the word "automated"
  2. Abstract: p1 line 20: remove word "such"
  3. Introduction: p1 line 37 rephrase "handling high numbers of imaging in minimum time" - it is unclear what "minimum time" means here. Probably ML will increase time efficiency or to increase the number of PET-CTs read by physicians. 
  4. M&M: p2 line 66 rephrase "histological proven" to "histology-confirmed".
  5. M&M: In general, the patient collective should be described in more detail. At least, a general statement on the PET-CT study question (initial staging, follow-up staging, ...?) should be made. 
  6. M&M: p. 3 line 93 rephrase "Mediso output"
  7. Results: Table 2 - please add the overall lesions sums to the table. 
  8. Results: overall very impressive and presented in a very comprehensible way. 
  9. Discussion: p10 line 234 "radiation exposure could be enhanced by fully diagnostic CTs..." - I would recommend to write "increased" instead of "enhanced"
  10. Discussion: p10 line 238-240: This paragraph seems like a overstatement - especially the advantages of application in other tumor entities cannot be stated based on the data presented by the authors. Yet, citation of corresponding published literature may still make this statement valid.
  11. Discussion: p10 line 254 "unnecessary side effects": please remove or rephrase the word "unnecessary" because there is nothing like "necessary side effects". 
  12.  Conclusions: ok.

Author Response

The authors present a very interesting and highly important analysis on ML utilization in hotspot classification of PSMA-PET-CT for prostate cancer. The manuscript is well-written.

> Thank you very much for this nice comment.

Despite its strengths, I recommend addressing the following comments:

  1. Title: The authors rely on a manual segmentation / did not apply a fully automated segmentation. Therefore, I would suggest to rephrase the title in a more cautious way. For example, I would recommend removing the word "automated" >  revised as requested (changed the title)
  2. Abstract: p1 line 20: remove word "such" > revised as requested (p1 line 22)
  3. Introduction: p1 line 37 rephrase "handling high numbers of imaging in minimum time" - it is unclear what "minimum time" means here. Probably ML will increase time efficiency or to increase the number of PET-CTs read by physicians. > revised as requested (p1 lines 40, 41)
  4. M&M: p2 line 66 rephrase "histological proven" to "histology-confirmed". > revised as requested (p2 line 70)
  5. M&M: In general, the patient collective should be described in more detail. At least, a general statement on the PET-CT study question (initial staging, follow-up staging, ...?) should be made. > We included additional patient information including the study question which was in all cases follow-up staging with questions for radium-223-dichloride treatment or lu-177-PSMA treatment. (p2 lines 72-77)
  6. M&M: p. 3 line 93 rephrase "Mediso output" > revised as requested (p3 lines 104, 105)
  7. Results: Table 2 - please add the overall lesions sums to the table. > revised as requested. Also the overall tables’ layouts have been improved.
  8. Results: overall very impressive and presented in a very comprehensible way. > thank you very much for this nice comment
  9. Discussion: p10 line 234 "radiation exposure could be enhanced by fully diagnostic CTs..." - I would recommend to write "increased" instead of "enhanced" > revised as requested (p9 line 256)
  10. Discussion: p10 line 238-240: This paragraph seems like a overstatement - especially the advantages of application in other tumor entities cannot be stated based on the data presented by the authors. Yet, citation of corresponding published literature may still make this statement valid. > we agree with the reviewer that this statement is too much and removed it at this position. Therefore, we included this at a later point as an outlook (p10 lines 292, 293).
  11. Discussion: p10 line 254 "unnecessary side effects": please remove or rephrase the word "unnecessary" because there is nothing like "necessary side effects". > revised as requested (p10 lines 278, 279)
  12.  Conclusions: ok.

Reviewer 2 Report

The authors here present data of machine learning models that reveal similar classification of pathological PSMA PET/CT findings compared to NM readers. The purpose is contemporary, since automated image reading is of high research interest.

Beside the small sized cohort (n=72) as a major limitation, there are some aspects to address: 

Abstract: appropriate

Introduction: clearly structured background and description of the objective

Materials and methods:

  • Please define the level of experience of the NM physicians
  • How was a pathological hotspot defined? Please indicate the characteristics and cut-offs (e.g. SUV max)
  • Authors state that patients were between 48 to 87 years. However, there is no further description of patient characteristics (number of treatment-naive patients or patients with biochemical recurrences, type of therapies, risk groups ...). It would be very interesting to get data on correlations with these subgroups
  • Why did you decide to choose a number of 48 patients for training and 24 patients for the validation set? Please explain and also state why 2 patients are missing in these sets. 
  • Why did you divide the cohort into 30 and 42 patients for inter-observer variability analysis? Please explain. 
  • Line 146: Again, please define "highly experienced NM physician"

Discussion:

  • Superiority of combined PSMA PET/CT compared to the single modalities is no new finding independent of ML and should not be highlighted as a results of this study
  • Low specificity is a major limitation because it leads to false treatment pathways (e.g. systemic therapy compared to local therapy). This should be highlighted

Author Response

The authors here present data of machine learning models that reveal similar classification of pathological PSMA PET/CT findings compared to NM readers. The purpose is contemporary, since automated image reading is of high research interest.

Beside the small sized cohort (n=72) as a major limitation, there are some aspects to address: 

> Thank you very much for the comments.

Abstract: appropriate

Introduction: clearly structured background and description of the objective

Materials and methods:

  • Please define the level of experience of the NM physicians > We have included this information into the manuscript (page 3 line 92 and section 2.4.)
  • How was a pathological hotspot defined? Please indicate the characteristics and cut-offs (e.g. SUV max) > This analysis was performed visually, no specific uptake threshold have been used. A hotspot was defined as uptake beyond the local background. This information was added to the manuscript. (page 3)
  • Authors state that patients were between 48 to 87 years. However, there is no further description of patient characteristics (number of treatment-naive patients or patients with biochemical recurrences, type of therapies, risk groups ...). It would be very interesting to get data on correlations with these subgroups > All patients were referred with the question of treatment with either radium-223-dichloride or Lu-177-PSMA. All patients had previous therapies which are now included in the manuscript (page 2, lines 72-77). We agree that analysis of the subgroups might be of interest, however, see it beyond the scope of this paper with the actual number of patients.
  • Why did you decide to choose a number of 48 patients for training and 24 patients for the validation set? Please explain and also state why 2 patients are missing in these sets. > As described in section 2.3, the whole data-set (72 patients in total) was sub-divided into training and validation sub-groups (including 48 and 24 patients respectively which add-up to 72) for cross-validation (CV). The validation set should be big enough (but smaller than the training cohort) and has to include a subset of the main cohort with similar characteristics as the training cohort. CV is applied to provide a measure of generalizable results. The hyperparameters are tuned on the training set, but tested on the hold-out. This strategy also reduces the probability of over-fitting.
  • Why did you divide the cohort into 30 and 42 patients for inter-observer variability analysis? Please explain. > due to availability of the retrospective data as well as different time limits, the NM physicians were assigned to annotate different numbers of patients. However, we made sure that the cohorts had similar demographic and physiological distributions. This information is added to the manuscript (page 5 lines 158-161).
  • Line 146: Again, please define "highly experienced NM physician" > as stated before, we included this information in the manuscript (page 3 line 92 and section 2.4.)

Discussion:

  • Superiority of combined PSMA PET/CT compared to the single modalities is no new finding independent of ML and should not be highlighted as a results of this study > We agree with the reviewer that in general the advantage of hybrid imaging modalities compared to single modalities is not new. However, in this case we could obtain significant information by the textural parameters obtainedin the low-dose CT without contrast media. So far, at least to our knowledge, it has not been shown before, that textural features based on low-dose CT data can improve lesion classification. Therefore we still highlighted this findings, but changed the text accordingly. (page 9 lines 246-252)
  • Low specificity is a major limitation because it leads to false treatment pathways (e.g. systemic therapy compared to local therapy). This should be highlighted > We changed the Discussion accordingly. (page 9 lines 252-254)

Reviewer 3 Report

The manuscript should be modified in several aspects; first of all, it requires an extensive english revision as well as an optimization of tables' layout.

Secondly, the paper should add further details in each of its section (e.g. the discussion is too short and synthetic).

The "hot topic" of machine learning nowadays is the differentiation between benign and malignant conditions/localizations, for this reason a "simply" discrimination of physiological uptakes from the pathological ones is too "weak"; furthermore, do the Authors want to create a sort of atlas of physiological uptakes to compare it with the uptake patterns of the patients (e.g. see the "Imiomics method" described by Strand et al, 2017)?

Page 1, line 40: the Authors wrote "CT" but the extended definition is three lines below (line 43); please correct it.

For these reasons, the manuscript in its current form is not ready for publication.

Author Response

The manuscript should be modified in several aspects; first of all, it requires an extensive English revision as well as an optimization of tables' layout. > as suggested, an extended English revision has been done by a native speaker. In addition, we improved the tables’ layouts. Specifically, a new column and a new row containing total numbers of hotspots are added to table 2. 

Secondly, the paper should add further details in each of its section (e.g. the discussion is too short and synthetic > We revised the whole manuscript and especially extended the Discussion).

The "hot topic" of machine learning nowadays is the differentiation between benign and malignant conditions/localizations, for this reason a "simply" discrimination of physiological uptakes from the pathological ones is too "weak"; furthermore, do the Authors want to create a sort of atlas of physiological uptakes to compare it with the uptake patterns of the patients (e.g. see the "Imiomics method" described by Strand et al, 2017)? > We agree with the reviewer, that this is the most important point. Indeed the manuscript was written a bit unclear. However, we classified the lesions as malignant versus others (physiological as well as benign or nonspecific uptake), as for later application of this method the malignant uptake is the relevant information. We changed the manuscript accordingly and hope it is now clearer. (e. g., page 1 line 19, page 4 line 117, page 8 line 224, etc.)

Page 1, line 40: the Authors wrote "CT" but the extended definition is three lines below (line 43); please correct it. > revised as requested (page 1 line 24, first time PET/CT appears)

For these reasons, the manuscript in its current form is not ready for publication.

Round 2

Reviewer 1 Report

The authors addressed the points mentioned comprehensively.

Author Response

Thank you very much for this nice comment.

Reviewer 3 Report

Dear Authors, 

 thanks for your extensive revision; please check line 297: "maligant" instead of "maligNant".

Apart from this minor correction, the manuscript is now ready for publication.

Author Response

thanks for your extensive revision; please check line 297: "maligant" instead of "maligNant". > revised as requested

Apart from this minor correction, the manuscript is now ready for publication. > Thank you very much for this nice comment.